

# The effects of anthropogenic disturbance and seasonality on the ant communities of Lang Tengah Island

Sze Huei Yek[1,2,*], Taneswarry Sethu Pathy[1,*], Deniece Yin Chia Yeo[1] and Jason Yew Seng Gan[1]

[1] School of Science, Monash University Malaysia, Bandar Sunway, Malaysia
[2] Institute for Tropical Biology & Conservation, Universiti Malaysia Sabah, Kota Kinabalu, Sabah, Malaysia
[*] These authors contributed equally to this work.

## ABSTRACT

Anthropogenic disturbances and seasonal changes significantly impact diversity and community composition of ants, but their effects are often intertwined. We investigated these drivers on Lang Tengah Island, a location with a pronounced monsoon season and three resorts that close during this period. We surveyed four sites, two disturbed and two undisturbed, before and after the monsoon season, using pitfall traps to sample epigaeic ant communities. Undisturbed habitats had higher species diversity, but both habitats (undisturbed and disturbed sites) have a high proportion of ants with characteristics of being encroached by generalist and invasive/tramp ant species. Post-monsoon sampling yielded an increase in species richness and diversity. Seasonal changes, such as monsoonal rains, can temporarily alter ant interactions and resource distribution, potentially maintaining diversity. Future studies should validate these findings for ant communities under similar pressures, using ant composition and functional roles for conservation and management purposes.

## INTRODUCTION

Ants (Hymenoptera: Formicidae) are globally distributed insects that play vital ecological roles as decomposers, soil aerators, seed dispersers, and key members of food webs (*Hölldobler & Wilson, 1990*). Ants are excellent ecological indicators due to their impact on resource availability through physical and chemical changes in their environment (*Jones, Lawton & Shachak, 1994*), ease of consistent sampling from their stationary colonies, and ecological importance (*Hölldobler & Wilson, 1990*). Changes in ant diversity, composition, and functional roles in response to environmental triggers have been extensively studied, particularly in restoration ecology (*Majer & Nichols, 1998*; *Bisevac & Majer, 1999*; *Casimiro, Sansevero & Queiroz, 2019*). Previous studies have explored ant responses to restoration efforts, including the effects of habitat fragmentation, invasive species, and changes in land use (*Majer & Nichols, 1998*; *Bisevac & Majer, 1999*; *Casimiro, Sansevero & Queiroz, 2019*).

Corresponding author
Sze Huei Yek, szehuei@ums.edu.my

Ant distributions and their community composition response to seasonality are well known (*Parr & Andersen, 2008*; *Linksvayer & Janssen, 2009*; *Gibb et al., 2015*; *Paolucci et al., 2017*; *Andersen, 2019*). The species diversity and community structure of epigaeic ants experience short-term changes due to natural disturbances, such as seasonal flooding (*Mertl, Ryder Wilkie & Traniello, 2009*; *Vasconcelos et al., 2010*). Some epigaeic ant species have adapted to these circumstances and are now more likely to survive (*Purcell et al., 2014*; *Kolay & Annagiri, 2015*; *Hertzog et al., 2016*), resulting in altered dynamics of the local population and species composition due to the coexistence of well-adapted and poorly adapted species (*Vasconcelos et al., 2010*).

Numerous studies have been done on how anthropogenic disturbances affect ant dispersal and community makeup (*Andersen, 1995*; *Andersen, 2019*; *Philpott et al., 2010*). Some studies have noted a loss in species richness because of human disturbance (*Yamaguchi, 2004*; *Lessard & Buddle, 2005*), while *Gibb & Hochuli (2003)* found identical species richness between disturbed and undisturbed areas. It is well recognised that urbanisation decreases species richness (*Lessard & Buddle, 2005*), allowing opportunistic or non-native species to persist, changing community composition (*Carpintero, Reyes-Lopez & De Reyna, 2003*; *Gibb & Hochuli, 2003*; *Holway & Suarez, 2006*). Exotic, invasive, and tramp ant species are frequently found in anthropogenically disturbed settings (*Brassard et al., 2021*) and these ant species have an adverse effect on native ant communities (*Human & Gordon, 1996*; *Holway et al., 2002*; *Holway, 2005*), non-ant invertebrate communities (*Gerlach, 2004*), and plant communities (*Ness & Bronstein, 2004*), ultimately compromising ecosystem services (*O'Dowd, Green & Lake, 2003*).

Ant community response to disturbance can be studied by classifying them into functional groups based on their broad behavior at biogeographical scales (*Hoffmann & Andersen, 2003*). This classification is sometimes preferred as it allows for easier comparison between studies and general understanding of disturbance impacts (*Andersen, 2019*). Seven functional groups are recommended for ant community comparison studies (*Andersen, 2000*), and they can be broadly categorized into generalist or specialist functional groups based on their behavior (*Hoffmann & Andersen, 2003*). The study of the effect of disturbance on ant communities commonly uses this functional group classification, as it is simpler to classify ants at higher taxonomic levels, such as genera and species-groups (*Andersen, 2019*).

Ant communities vary in functional group composition across different habitats, with dominant generalist functional groups such as Dominant Dolichoderinae (DD), Generalized Myrmicinae (GM), and Opportunists (OPP) being more common than specialist functional groups such as Hot Climate Specialists (HCS), Specialist Predator (SP) and Cryptic Species (CRY) (*Leal et al., 2012*; *Arnan et al., 2018*; *Triyogo et al., 2020*). The original functional group classification was based on Australian forest ant communities (*Andersen, 1995*), it has been adapted for other landscapes, including tropical forests in Brazil and Indonesia (*Leal et al., 2012*; *Triyogo et al., 2020*). Regardless of the habitat type, generalist functional groups typically make up more than 80% of the ant community, with the remainder being specialist functional groups. These findings were from grassland and agroforest landscapes in Australia and Indonesia (*Andersen, 1995*; *Andersen, 2000*;

*Triyogo et al., 2020*). In pristine forest fragments in Australia, Brazil, Western United States and Spain, a higher proportion of specialist functional groups ($\sim$38%) than those in anthropogenically disturbed environments were recorded (*Leal et al., 2012*; *Carpintero, Reyes-Lopez & De Reyna, 2003*; *Gibb & Hochuli, 2003*; *Holway & Suarez, 2006*), indicating a permanent alteration of ants composition in disturbed landscapes.

With increasing anthropogenic presence globally, it is imperative to understand how different types of disturbances affect species richness, community composition of ants to prevent further biodiversity loss (*Allen, Ewel & Jack, 2001*; *Luck, 2007*). This study aims to understand the impact of anthropogenic disturbance and seasonality on the epigaeic ant fauna of Lang Tengah Island, located off the East Coast of Peninsular Malaysia. The study focused on comparing species richness, diversity indices and composition of ants in disturbed and undisturbed habitats, as well as before and after the monsoon season. We hypothesize that monsoonal rains will lead to short-term changes in species composition due to shuffling of niches (*Mertl, Ryder Wilkie & Traniello, 2009*; *Vasconcelos et al., 2010*). Furthermore, disturbed habitats will see a decrease in species richness and diversity (*Yamaguchi, 2004*; *Lessard & Buddle, 2005*). This is due to habitat changes favouring more generalist and non-native ant species at these human-modified landscapes (*Brassard et al., 2021*).

## MATERIALS & METHODS
### Study area
This study was conducted at four sites on Lang Tengah Island, which is located in the state of Terengganu off the east coast of Peninsular Malaysia and has an area of approximately 125 hectares. There are three resorts on the island (Sari Pacifica, Summer Bay, and D'Coconut). Sari Pacifica and Summer Bay are located adjacent to each other. However, we did not have permission to conduct sampling activities at Summer Bay resort. To examine the effect of anthropogenic disturbance, we selected all the sites based on their accessibility by foot. We designated Sari Pacifica resort (SP, 5°47′32.64″N 102°53′36.09″E) and D'Coconut resort (DC, 5°47′22.56″N 102°53′56.21″E) as anthropogenically disturbed sites due to resort construction, resulting in landscape changes and human presence. The vegetation at these disturbed sites consist of ornamental shrubs such as *Hibiscus* hybrids, Ixora *spp.* (*Gan et al., 2022*). With the land clearing for anthropogenic construction, disturbed sites are devoid of tall canopy trees. Batu Kuching (BK, 5°47′21.73″N 102°54′8.31″E) and Turtle Bay forest (TB, 5°47′22.34″N 102°54′1.66″E) were designated as undisturbed sites due to their relatively pristine conditions and limited accessibility to tourists. These sites have hiking trails leading to lookout points on the island and consist of relatively open forest habitat (Fig. 1; *Google Maps, 2022*).

The size of each site is within the range of 53 m². The distance between the disturbed sites is 690 m, whereas the distance between the two undisturbed sites is about 200 m. These sites were chosen due to accessibility as only the western part of the island and a few southern parts are accessible, while the eastern part of the island consists of rocky outcrops with no anthropogenic activities present except for occasional hiking trails scattered around this area (Fig. 1; *Google Maps, 2022*).

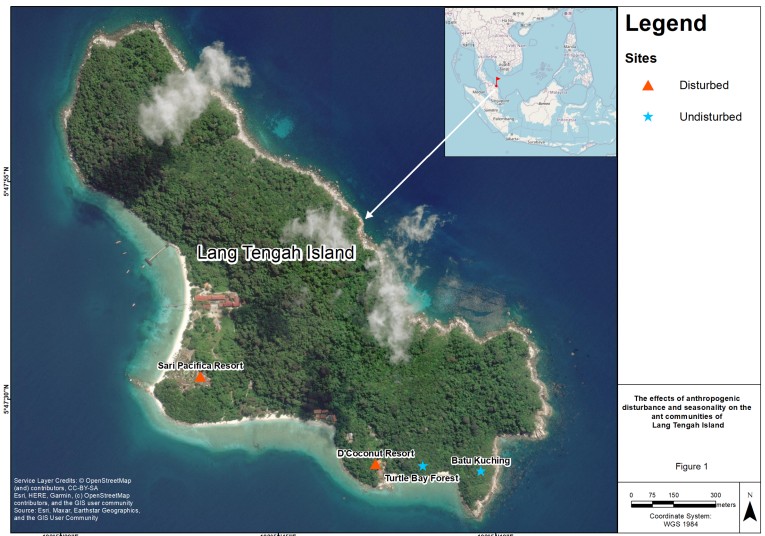

**Figure 1 Map of four sampling sites on Lang Tengah island.** Lang Tengah island is located on the east coast of Peninsular Malaysia. Sari Pacifica (SP) resort and D'Coconut (DC) resort were designated as disturbed sites while Batu Kuching (BK) and Turtle Bay forest (TB) were designated as undisturbed sites. Map data @ 2022 Imagery @ 2022 CNES/Airbus, Landsat/Copernicus, Maxar Technologies.

Field experiments were approved by two Resort Operators, namely Mr. Steve from Sari Pacifica Resort and Mr. Daniel from D'Coconut Resort. Additionally, field experiments were also approved by Principal Officer (Dr. Long Seh Ling) from Lang Tengah Turtle Watch (NGO) on Lang Tengah Island. This island experiences a pronounced monsoon period, resulting in dry and rainy seasons. There are no indigenous human settlements on this island. Due to the harsh weather during the monsoon period, the island is closed to tourists for six months, and all resort staff are evacuated from October to March. To examine the effect of seasonality, we sampled the four sites twice, once post-monsoon from 2nd to 7th March 2020, right after the island reopened to tourists, and pre-monsoon sampling was carried out from 26th September to 1st October 2020, before the island was closed for the monsoon period. On the first day of sampling, we selected the sites and set up the pitfall traps. Pitfall traps were kept closed for 24 h before opening to reduce digging in effects. The traps were left open in the field for five days. On the 6th day of setting out traps, we dug up traps and packed them for transportation to the mainland. Although the pre-monsoon sampling was carried out during the early period of the global COVID-19 pandemic, daily cases in Malaysia were low during this period (*DG of Health, 2020*), and the government had just relaxed domestic travel restrictions, contributing to a period of heightened tourist activity during this time (*BERNAMA, 2020a*; *BERNAMA, 2020b*).

## Sampling of the epigaeic ant fauna

Ant diversity studies typically focus on three different vertical strata: canopy, arboreal and epigaeic or ground-dwelling (*Hashimoto et al., 2006*). While canopy and epigaeic ant communities have been found to show little overlap (*Itino & Yamane, 1995*; *Brühl,*

*Gunsalam & Linsenmair, 1998*; *Yanoviak & Kaspari, 2000*), arboreal ants can often be detected using epigaeic sampling methods as they come down to the ground to forage (*Wilson, 1959*; *Hahn & Wheeler, 2002*). In this study, due to the absence of canopy trees at the disturbed sites, an epigaeic sampling method was used to study the diversity of ground-dwelling ants. Adding specific arboreal sampling methods such as baits placed on trees may increase ant diversity and community composition. However, this can introduce sampling bias and lead to inconsistencies in the representation of ant communities, as not all arboreal ants are attracted to the same baits.

For each study site and pre-/post-monsoonal season sampling, we installed traps to collect the epigaeic ant fauna. We used pitfall traps adapted from *Hashimoto (2003)* with 200 ml of 75% ethanol. A total of 20 traps were installed at each site in a straight row, spaced about 1 m apart, resulting in 160 sampling points (4 sites × 20 repetitions × 2 pre-/post monsoonal seasons). *Eguchi, Bui & Yamane (2004)* found that the majority of epigaeic ants' foraging range does not exceed one meter, although some predatory raiding ants such as *Leptogenys* spp. might have a larger foraging range (*Maschwitz et al., 1989*). We chose the distance between the traps as 1 m because the ground surface at the resort sites was mostly concrete, limiting suitable trap placement to available ground surfaces consisting of soil and vegetation. This spacing was also used at the undisturbed sites for consistency, even though there was more surface area for trap placement. Most ant diversity studies using pitfall traps typically use a spacing range of five to 10 m (*Yek et al., 2009*).

The traps were kept in the field for five days and checked daily for any evaporation of the 75% ethanol. After five days, the traps were removed, and securely packed in waterproof bags for transport to the Monash University Malaysia laboratory for subsequent sorting. In the laboratory, the sorted ants were identified to the genus level following *Hashimoto (2003)* and *Bolton (1994)* by two of the authors (TSP and SHY). Species identification was carried out by comparing the specimens with the ant collections at the Institute for Tropical Biology and Conservation (ITBC), Universiti Malaysia Sabah (UMS). Morphospecies codes were assigned to unidentified species, which apply only to this study. A full set of specimens with voucher specimen numbers HYM0003796–HYM0003826 is deposited at ITBC, UMS (Table S1).

We analyzed the changes in ant composition, including their community identity and functional groups. To ensure accuracy, we used several databases to categorize the ant species. We first checked the Global Invasive Species Database (*GISD, 2023*) and the IUCN worst 100 list (*Lowe et al., 2000*) and then cross-referenced each identified species with AntWiki (*Janicki et al., 2016*; *Guénard et al., 2017*) for distributional and ecological information. The final assignment followed the terminology from AntWiki for native, tramp, and invasive species. Tramp status was used for species with uncertain biogeographic origins, but that have established populations in non-native habitats (*Brassard et al., 2021*).

The ant species were classified into seven functional groups based on *Andersen (2000)*. Four generalist functional groups exhibit broad foraging behaviour, while three specialist functional groups exhibit narrow foraging behaviour (*Andersen, 2000*; *Brandão, Silva & Delabie, 2012*). The generalist functional groups are Dominant Dolichoderinae (DD), Generalized Myrmicinae (GM), Opportunists (OPP) and Subordinate Camponotini (SC).

The specialist functional groups are Cryptic Species (CRY), Hot Climate Specialists (HCS) and Specialist Predators (SP).

## Data analyses

To examine sample coverage completeness, rarefaction curves and species richness estimators (Chao1 and ACE) were computed. The *iNext* and *vegan* package (*Chao, Chiu & Jost, 2014*; *Hsieh, Ma & Chao, 2020*) in R (v4.1.1; *R Core Team, 2022*) were used to calculate the Hill numbers, species richness estimators, and rarefaction curves. The study first assessed ant species richness using Hill numbers, which included species richness, Shannon diversity and Simpson diversity (*Colwell et al., 2012*). To represent a site/season, 20 traps were combined due to their close spacing (1 meter). Ant species incidence frequencies were used as input to compute Hill numbers, and species occurrences per sample were used as proxy for relative ant abundance to avoid bias from sampling near nests and trails (*Longino, 2000*; *Yek et al., 2009*).

To compare species composition between habitat types and seasonality, Jaccard similarity index was computed using *EstimateS* (v9; *Colwell, 2013*).

## RESULTS

A total of 2,084 individual ants, identified to 30 morphospecies, were collected at four sampling sites across pre-and post-monsoon seasons (Table S1). The species diversity estimators (Chao1 and ACE; Table 1) and rarefaction curves (Fig. S1) indicate that all sites were generally adequately sampled. Post-monsoon sampling of undisturbed sites yielded the highest species richness (Observed = 23 species, Estimated = 24–26 species) while the pre-monsoon sampling of disturbed habitats yielded the lowest species richness (Observed = 9 species, Estimated = 12–13 species) (Table 1). When assessing the species richness from each site, Turtle Bay Forest—an undisturbed site, yielded the highest species richness pre-monsoon (Observed = 17 species, Estimated = 21 species), whereas D'Coconut Resort—a disturbed site, yielded the lowest species richness post-monsoon (Observed = 5 species, Estimated = 6 species) (Table S2). In general, undisturbed sites (Turtle Bay Forest and Batu Kuching) presented higher species richness and diversity indices compared to disturbed sites (Sari Pacifica Resort and D'Coconut Resort). Moreover, post-monsoon at both disturbed and undisturbed sites also yielded an increase in species richness and diversity (Table 1).

From the data shown in Table 2, we can see that the species composition, using the Jaccard similarity index, only shares 10 to 20% similarity between disturbed and undisturbed sites. The species composition between monsoon shares 50 to 60% similarity from each other (Table 2).

We assigned epigaeic ants to their community identity status. Most of the identification was carried out at the morphospecies level, hence we conservatively assigned the unidentified species to native species (24 native species; Table 3). Six ant species were either tramp or invasive species (Table 3). At undisturbed sites, 25% of ant species were categorized as tramp/invasive species. At disturbed sites, 17% of ant species were categorized as tramp/invasive species (Fig. 2).

**Table 1  The results of species richness, Shannon diversity and Simpson diversity index on habitat types (undisturbed *versus* disturbed) and seasonality (pre-/post-monsoon).** Two species richness estimators (Chao1 and ACE) were computed to examine the completeness of the sampling effort.

| Habitat types | Disturbed | | Undisturbed | |
|---|---|---|---|---|
| Seasonality | Pre-monsoon | Post-monsoon | Pre-monsoon | Post-monsoon |
| Species richness | 9 | 11 | 13 | 23 |
| Shannon index | 1.12 | 1.47 | 2.18 | 2.00 |
| Simpson index | 0.54 | 0.71 | 0.87 | 0.80 |
| Chao1 estimate | 12 | 11 | 13 | 24 |
| SE Chao1 | 4.48 | 1.28 | 0.24 | 1.84 |
| ACE estimator | 13 | 13 | 13 | 26 |
| SE ACE | 1.46 | 1.68 | 1.67 | 2.44 |

**Table 2  Jaccard similarity index between sites and pre-/post-monsoon.** Disturbed and undisturbed sites shares 10 to 20% similarity in species composition, whereas pre- and post-monsoon shares 50 to 60% similarity in species composition.

| | Community | | Jaccard similarity index |
|---|---|---|---|
| | Sample 1 | Sample 2 | |
| Habitat types | Disturbed-pre monsoon | Undisturbed-pre monsoon | 0.1 |
| Habitat types | Disturbed-post monsoon | Undisturbed-post monsoon | 0.2 |
| Monsoon | Pre monsoon-disturbed | Post monsoon-disturbed | 0.5 |
| Monsoon | Pre monsoon-undisturbed | Post monsoon-undisturbed | 0.6 |
| Others | Post monsoon-disturbed | Pre monsoon-undisturbed | 0.2 |
| Others | Post monsoon-undisturbed | Pre monsoon-disturbed | 0.2 |

Ant communities were categorized into four generalist and three specialist functional groups (Table 3). Both disturbed and undisturbed sites have relatively high proportions of generalist species, making up to >80% of the ant community (Fig. 3). Generalized Myrmicinae (GM) and Opportunist (OPP) dominated in the disturbed sites whereas Subordinate Camponotini (SC) generalist groups were only found in undisturbed sites (Table 3). The relative proportion of generalist groups increased slightly post-monsoon for both disturbed and undisturbed sites (Fig. 3).

## DISCUSSION

The results of our study indicate that undisturbed habitats have higher ant species richness (Table 1), which is consistent with most ant bioindicator studies (*Uno, Cotton & Philpott, 2010*; *Buczkowski & Richmond, 2012*). This is most probably due to higher niche division between the ant species, hence allowing higher species richness to be harboured. On the other hand, disturbed sites have more generalist functional groups, as reported in other studies (*García-Martínez et al., 2015*), whereas specialist ant groups were more sensitive to anthropogenic disturbance (*Leal et al., 2012*). It is unlikely that the distance between the disturbed and undisturbed sites plays a crucial role as the distance between these sites are within the range of dispersal of these ants, which range from a few meters to around 30

**Table 3   Incidence of epigaeic ant morphospecies at each habitat type across pre-/post-monsoon season.**

| Status | Behaviour functional group | Species | Habitat type | | | |
|---|---|---|---|---|---|---|
| | | | Disturbed | | Undisturbed | |
| | | | Pre monsoon | Post monsoon | Pre monsoon | Post monsoon |
| Native | Generalist (DD) | *Dolichoderus* sp. | 0 | 1 | 0 | 7 |
| Tramp | | *Monomorium floricola* | 0 | 2 | 0 | 0 |
| Invasive | | *Monomorium pharaonis* | 6 | 13 | 0 | 0 |
| Native | | *Monomorium* sp. I | 12 | 10 | 0 | 1 |
| Native | Generalist (GM) | *Pheidole* sp. I | 0 | 1 | 0 | 0 |
| Native | | *Pheidole* sp. II | 2 | 11 | 0 | 0 |
| Native | | *Pheidole* sp. III | 5 | 18 | 0 | 0 |
| Native | | *Pheidole* sp. IV | 0 | 1 | 0 | 0 |
| Invasive | Generalist (OPP) | *Anoplolepis gracilipes* | 9 | 4 | 0 | 0 |
| Native | | *Aphaenogaster* sp. | 0 | 1 | 0 | 0 |
| Native | | *Cardiocondyla kagutsuchi* | 0 | 1 | 0 | 0 |
| Native | | *Cardiocondyla* sp. | 4 | 0 | 0 | 0 |
| Native | | *Cardiocondyla tjibodana* | 2 | 1 | 0 | 0 |
| Native | | *Paraparatrechina* sp. | 5 | 13 | 0 | 0 |
| Invasive | | *Paratrechina longicornis* | 3 | 2 | 20 | 34 |
| Native | | *Paratrechina* sp. | 0 | 1 | 0 | 0 |
| Tramp | | *Tapinoma indicum* | 16 | 28 | 0 | 0 |
| Invasive | | *Tetramorium insolens* | 0 | 2 | 6 | 9 |
| Native | | *Tetramorium* sp. I | 3 | 2 | 10 | 8 |
| Native | | *Tetramorium* sp. II | 0 | 0 | 0 | 5 |
| Native | | *Tetramorium* sp. III | 0 | 0 | 12 | 17 |
| Native | Generalist (SC) | *Camponotus* sp. | 0 | 0 | 1 | 1 |
| Native | Specialist (HCS) | *Meranoplus bicolor* | 15 | 8 | 0 | 0 |
| Native | Specialist (SP) | *Amblyopone* sp. | 0 | 0 | 1 | 1 |
| Native | | *Leptogenys* sp. | 0 | 0 | 4 | 0 |
| Native | Specialist (CRY) | *Carebara* sp. I | 0 | 0 | 1 | 3 |
| Native | | *Carebara* sp. II | 0 | 1 | 0 | 0 |
| Native | | *Hypoponera* sp. I | 1 | 3 | 0 | 0 |
| Native | | *Hypoponera* sp. II | 0 | 1 | 0 | 0 |
| Native | | *Plagiolepis* sp. | 0 | 3 | 3 | 0 |

**Notes.**

The ants are grouped to their status (column 1) following *Global Invasive Species Database (2023)*, IUCN worst 100 list (*Lowe et al., 2000*) and finally checked against the map distribution in AntWiki (*Janicki et al., 2016*; *Guénard et al., 2017*).

Functional groups assignment is further classified into generalist and specialist functional groups with the code as follows: DD, Dominant Dolichoderinae; GM, Generalized Myrmicinae; OPP, Opportunist; SC, Subordinate Camponotini; CRY, Cryptic Species; HCS, Hot Climate Species; SP, Specialist Predator.
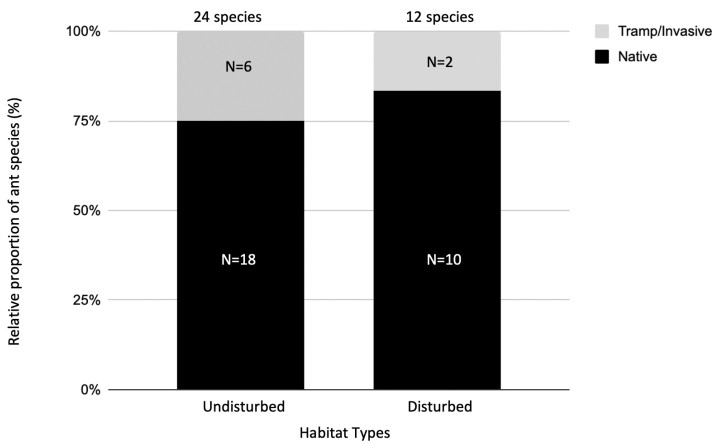

**Figure 2** **The relative proportion of native *versus* tramp/invasive ant species across undisturbed and disturbed habitat types.** There are 24 ant species in the undisturbed habitat sites and 12 ant species in the disturbed habitat sites. At undisturbed sites, 25% of ant species found here are categorized as tramp/invasive species. At disturbed sites, 17% of ant species found here are categorized as tramp/invasive species.

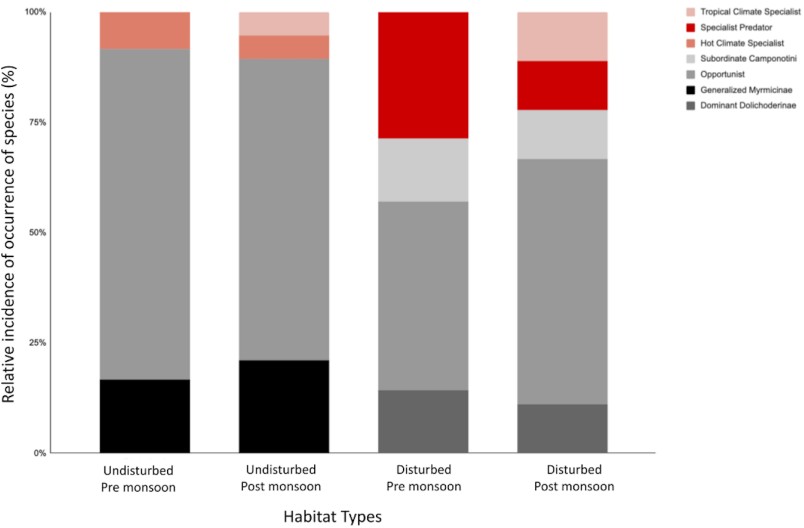

**Figure 3** **The relative incidence of epigaeic ant morphospecies classified into functional groups for habitat type and pre-/post-monsoon season.** The top three red-scale shaded bars represent specialist functional groups—Cryptic Species (CS), Hot Climate Specialists (HCS) and Specialist Predators (SP), and the bottom grey-scale shaded bars represents the generalist functional groups—Opportunists (OPP), Generalized Myrmicinae (GM), Dominant Dolichoderinae (DD) and Subordinate Camponotini (SC).

km from their natal nest, depending on the queen's body size (*Helms IV, 2018*). We believe that the findings of tramp/invasive and generalist ant species in both habitats reflect the altered landscape that favours these species. Abundant anthropogenic resources, such as food waste and relaxed resource competition may support an increased number of tramp and generalist ant species, but decrease the number of native and specialist ant species (*Rocha & Fellowes, 2020*).
Anthropogenic activities can lead to an increase in invasive and tramp ant species due to the creation of new ecological niches that these ants can exploit (*Shimoji et al., 2022*). In small islands, as in many urban environments, the constant influx of merchandise and human traffic through the jetty can make them an ideal gateway for exotic species introduction (*Russell et al., 2017*). Another possible transmission route for exotic species is through marine debris, which accumulates on the beaches, particularly after the monsoon season (*Fauziah et al., 2021*; *Tan et al., 2022*; Fig. 1). Although regular beach clean-up activities are carried out during the tourist seasons by workers of the resorts or volunteers from NGOs (*Yi & Kannan, 2016*), many tramp/invasive ant species could still occupy undisturbed sites. Because there are no distinct dispersal barriers for ants to spread (rivers, lakes or mountains) on the island, it is likely that natural agents such as winds or anthropogenic agents can transport ants between disturbed and undisturbed areas. Regardless of the means of transport, the short sampling period, small sample size, dominance and presence on the undisturbed sites could explain the overall low species richness of epigaeic ants sampled at Lang Tengah Island.

The annual monsoonal rains, in short-terms, may introduce heterogeneity to the environment through microclimate characteristics (*Bátori et al., 2019*), habitat structure (*Lassau & Hochuli, 2004*), resource availability (*Souza et al., 2015*; *Silveira et al., 2016*) and competitive interactions (*Rowles & O'Dowd, 2007*), which may have caused the 50–60% changes in *Jaccard* similarity index between the season. The impact of seasonality, such as the monsoonal rains, can significantly affect epigaeic ant abundance by causing nest flooding and increasing mortality rates (*Mertl, Ryder Wilkie & Traniello, 2009*; *LeBrun, Moffett & Holway, 2011*). Indirectly, it can reset ant-ant interactions for resource utilization, leading to short-term changes in species composition (*Gray et al., 2018*), which is consistent with the observed changes in functional group composition.

In both disturbed and undisturbed habitats, generalist species dominate over specialist species due to their ability to flexibly exploit different resources and expand their colonies rapidly, leading to their ecological dominance (*Brandão, Silva & Delabie, 2012*). Generalist species can also coexist with other generalist species by exploiting different unoccupied niches (*Parr & Gibb, 2012*). Although the composition and identity of these groups of ants remained largely unchanged, large-scale monsoonal rains caused a slight increase in their abundance in both habitats (*Tchoudjin et al., 2020*).

## CONCLUSIONS

In conclusion, our study revealed two ant composition patterns on Lang Tengah: (1) prevalent tramp/invasive at both habitat types, higher at undisturbed sites due to the constant flux of new propagules through marine debris and anthropogenic activity encroaching into these sites, and (2) seasonality such as monsoonal rains resets ant-ant interactions and resources distribution in the short-term, potentially maintaining diversity. Future studies should validate the findings of this study across different ant communities under similar pressure. We propose the use of ant composition status and functional roles in future studies as these categorizations are more relevant for biodiversity conservation

and management plans than the species richness/diversity indices. For validation of these results, we propose longer time-based sampling periods pre- and post-monsoon to better reveal the community and functional dynamics on ants on islands.

## ACKNOWLEDGEMENTS

We are grateful to the members of the Terrestrial Ecology Laboratory, Monash University Malaysia for assistance in data collection during the sampling seasons. We thank many reviewers for their suggestions to improve the data interpretation of this manuscript.

### Funding

The funding was provided by Monash University Malaysia Honours fellowships to Taneswarry Sethu Pathy (2020) and Seed Grant to Sze Huei Yek (2018–2020). The funders had no role in study design, data collection and analysis, decision to publish, or preparation of the manuscript.

### Grant Disclosures

The following grant information was disclosed by the authors:
Monash University Malaysia Honours fellowships to Taneswarry Sethu Pathy (2020).
Sze Huei Yek (2018–2020).

### Competing Interests

The authors declare there are no competing interests.

### Author Contributions

- Sze Huei Yek conceived and designed the experiments, analyzed the data, prepared figures and/or tables, authored or reviewed drafts of the article, and approved the final draft.
- Taneswarry Sethu Pathy conceived and designed the experiments, performed the experiments, analyzed the data, authored or reviewed drafts of the article, and approved the final draft.
- Deniece Yin Chia Yeo performed the experiments, authored or reviewed drafts of the article, and approved the final draft.
- Jason Yew Seng Gan performed the experiments, prepared figures and/or tables, authored or reviewed drafts of the article, and approved the final draft.

### Field Study Permissions

The following information was supplied relating to field study approvals (*i.e.*, approving body and any reference numbers):

Field experiments were approved by two Resorts Operators, namely Mr. Steve from Sari Pacifica Resort and Mr. Daniel from D'Coconut Resort. Additionally, field experiments was also approved by Principal Officer (Dr. Long Seh Ling) from Lang Tengah Turtle Watch (NGO) on Lang Tengah Islands.

## Data Availability

The raw measurements are available in the Supplemental Files.

## Supplemental Information

Supplemental information for this article can be found online at http://dx.doi.org/10.7717/peerj.16157#supplemental-information.

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
