# Peer review of "The effects of anthropogenic disturbance and seasonality on the ant communities of Lang Tengah Island"

_PeerJ, doi:10.7717/peerj.16157_

## Round 0.1 · original submission · Major Revisions

Dear Authors

Thank you for submitting to PeerJ. The manuscript has been evaluated and found major revisions. See the comments of the reviewer, particularly reviewer 1 to consider the changes.

·

Basic reporting

No comment.

Experimental design

The sample size is extremely small (of which authors provided satisfactory justification), I think the study is worth publishing. However, because of the small sample size, I would recommend that the study be considered for publication as a short communication or a checklist rather than a full article. However, I am not sure if PeerJ publishes short communications or checklists (that can also be an alternative), when I checked online I did not see something that speaks to short communication. I think the manuscript can benefit from statistical tests that compare species richness, diversity indices and composition between disturbed and undisturbed, as well as before and after monsoon season. However, I am aware that the sample size is going to be a limitation when it comes to many statistical tests but I made recommendations in the data analyses section.

Validity of the findings

I think if authors can revise this manuscript and use statistical tests, that would strengthen the results. This would therefore improve the discussion also.

Additional comments

The authors present an interesting study that is looking at the effect of anthropogenic disturbance and seasonality on ant communities of Lang Tengah Island. Despite the extremely small sample size (of which authors provided satisfactory justification), I think the study is worth publishing. However, because of the small sample size, I would recommend that the study be considered for publication as a short communication or a checklist rather than a full article. However, I am not sure if PeerJ publishes short communications or checklists (that can also be an alternative), when I checked online I did not see something that speaks to short communication. I think the manuscript can benefit from statistical tests that compare species richness, diversity indices and composition between disturbed and undisturbed, as well as before and after monsoon season. However, I am aware that the sample size is going to be a limitation when it comes to many statistical tests but I made recommendations in the data analyses section.
Below are comments to specific sections of the manuscript.

Abstract:
What did you find when comparing ant diversity and communities before and after monsoon season? Can you say something about the communities of ants in disturbed and undisturbed sites? A sample size of four sites is rather very small, however I understand the nature of the island – can this be considered as a short communication rather?
Line 24: add composition “of ants”
Lines 36 to 37: I suggest you remove island ant composition, pitfall traps and rarefication curves.

Introduction
Line 44: “Ants are excellent...”, instead of “They are excellent...”
Line 64: “while others did not observe such decline”, can the authors be specific on what Gibb and Hochuli (2003) found, like did they record an increase in species richness in disturbed habitats compared to undisturbed or did they find similarities between disturbed and undisturbed habitats? Also, remove “others” since they cited just one study.
Line 87: Change “While” to “However”
Line 93: Leal et al. (2012) recorded high proportion of specialist functional groups compared to what?
Lines 101 to 104: I suggest you rephrase that last sentence, and remove “The study uses pitfall trapping to collect data and analyse.” Maybe say “The study focused on comparing species richness, diversity indices and composition of ants in disturbed and undisturbed habitats, as well as before and after the monsoon season.” Instead of saying species richness, diversity indices and composition of functional groups and status, because to me this reads as if you ran your analyses for each functional group or status – although that would be great I do not think your sample size would allow. I am suggesting removing rarefaction curves because these show whether your sampling was adequate or not rather than telling us much about the effect of disturbance or season on ant composition. I suggest that the authors provide hypotheses that are supported by literature after their aims.

Materials and Methods
Data were collected from four sites, as I mentioned already, that is a very small sample size. However, in Lines 111 to 115 the authors provided the justification for their sample size, which could not be avoided.
Line 110: mentioned the three resorts on the island; Sari Pacifica, Summerbay and D’Coconut. Then Line 117 talks about “Sari Pacifica-Summerbay resort and D’Coconut resort being disturbed sites”. Now I am not sure of the reason for writing “Sari Pacifica-Summerbay resort” as a pair, clarity is needed here. Authors need to indicate how far apart were the sites within the disturbed habitat, as well as within undisturbed, also distance between disturbed and undisturbed habitats. Additionally, can they provide the size of each site.
Line 140: Change “Ants’ diversity” to “Ant diversity.”
Line: 145: Talks about absence of canopy trees at disturbed sites. Can I suggest that in the previous paragraph authors provide a brief description of vegetation type at the disturbed sites? At least in undisturbed habitats there was a mention of a forest.
Lines 129 to 131: Indicates that sampling occurred from 2 to 9 March 2020, as well as 26 September to 3 October 2020. Now, line 163 says traps were in the field for five days. This five-day period contradicts the seven days I counted when I was looking at the sampling dates provided in Lines 129 to 131.
Line 167: Abbreviations need to be defined before use. It took me some time to realize that TSP and SHY are authors, may I suggest that the authors write “... by two of the authors (TSP and SHY)”. In this way you are helping your reader know what do TSP and SHY stand for.
Given that in the aims of the study the authors talked about functional groups, I am missing how species or morphospecies were assigned to groups or the criterion used, I would have expected this to be talked about before the data analyses section.
Line 13 change “analysis” to “analyses.”
Data analyses
Although I understand that the authors had a smaller sample size, as a result they are limited on statistical methods to use. Can I suggest that they support the output of frequencies by doing a chi-squared test for species richness, Shannon and Simpson indices. Chi-squared test will provide p-values, of which those weigh more compared to just comparing values. In addition, can I suggest that they calculate the Jaccard index of similarity to compare species composition between disturbed and undisturbed, as well as between before and after the monsoon season. If they have other alternative statistical tests that can work with their sample size, those can be used also. I suggest that Lines 179 to 182 be moved to start this data analyses section.
Lines 183 to 196, I suggest be moved to before the data analyses section, this is what I was saying it is missing – assignment of species to functional groups.
Results
Line 200: Supplementary Table S1 was not part of the document I reviewed.
Line 201: Can you remind your reader on what do TB and BK stand for.
Line 201: Change “diversity indexes” to “diversity indices.”
Lines 201 to 203: I feel these results can be strengthened by statistical tests rather than just relying on frequencies. For instance, statistically there may be no differences between 9 and 13. “Post-monsoon at disturbed sites yielded an increase in species richness and diversity but not at undisturbed sites (Table 1).” I am not following here, in table 1, richness in disturbed sites was 9 before and 11 after monsoon, while in undisturbed it was 13 before and 23 after monsoon – so, how is it that the increase was not observed at undisturbed sites?
Lines 207 to 215: I suggest that the estimated species richness be presented before talking about the results on disturbance and season. Meaning move this to the beginning of the results section. Part of this feels like repetition because it is what is talked about already when you were reporting the species richness.
Line 212: “Sampling unit” is a new term, I do not remember it being mentioned before. If authors were referring to sites, I suggest they use site that for consistency.
Line 215: I do not have “Supplementary Table S2.”
Lines 222 to 223: Replace “found here are” with “were”
Line 229: “dominated” instead of “dominate”
Line 230: “were” instead of “are”
This section on ant communities can be strengthened by the Jaccard index of similarity that I suggest or any statistical test for composition.

Discussion
Line 241: Phrase as “This study assessed changes in epigaeic ant fauna in response....”
Lines 242 to 244: Says there was higher species richness in undisturbed habitats and before the monsoon – I think this contradicts what is presented in Table 1 when focusing on the season.
Lines 242 to 250 is repeating the results, rather than discussing them. Authors need to remove citation of figures and tables in the discussion because these are cited in the results.
Lines 258 to 260: Can the distance between disturbed and disturbed habitats be a factor? And what about the dispersal abilities of these ants? My point on the distance and the dispersal ability still applies on the argument in Lines 267 to 269.
Line 276: Talks about changes in species composition observed and Table 1 and Figure 4 are cited, and I do not see anything that speaks to species composition in Table 1. Also this contradicts what is said in Lines 286 to 288 – “Although the composition and identity of these groups of ants remained largely unchanged...”
Lines 269 to 272: Maybe the short sampling period and small sample size could be a factors.

References
Sometimes authors use “et al.” when citing but sometimes they list all the authors. I thought they list all the authors when it is three authors but in some cases, e.g. Lines: 57, 181, 193 and 278 (Mertl et a. 2009; Hsieh et al. 2020; Brandao et al. 2012; LeBrun et al. 2011) I noticed that authors used “et al.” even though these are three authored papers. Authors need to be consistent.
Line 186: Lowe et al. 2004 not on the reference list.

·

Basic reporting

The article was written well. However, some sentences were rather long and could potentially confuse readers. I have suggested a few changes to the tenses in some sections.
Presentation of the results is good.

Experimental design

Although the design and repetitions were adequate, I felt that the number of sites was low. Thus, there could have been pseudo-replication. How far apart were the sites?

Validity of the findings

No comment.

---

## Round 0.2 · Minor Revisions

Dear Authors

Thank you for submitting the revised version. Some queries and suggestions are still from the reviewer. Please incorporate them in the revised version for the next decision. Comments are attached herewith.

·

Basic reporting

Clarity is provided on the revised version. There are a few things that require clarity still but generally the paper reads well.

Experimental design

The issue of small sample size remains, however, these are important data to get out and authors could not have done otherwise to increase it.

Validity of the findings

Given that the study was conducted in an island with these extreme seasonal changes, these findings will be a contribution to the biodiversity conservation community.

Additional comments

Review – 14 July 2023
The effects of anthropogenic disturbance and seasonality on ant communities of Lang Tengah Island
I wish to thank the authors for revising the paper. Despite the small sample size that I had raised in my previous revision, I think this work is worth publishing. So, I recommend acceptance with minor revision. Below I have a few comments.

Abstract:
Can you say something reporting the results based on season (e.g. high diversity post-monsoonal than pre-monsoonal). Maybe you can this sentence just before the sentence in Line 30 “Seasonal changes, such as monsoonal rains....”.

Introduction
Line 75: “Ant community response...”, instead of “Ant communities’s response...”
Line 81: Change “studies” to “the study” – because you cited a single study.
Line 92 to 93: Can you specify the region/country where such a percentage of generalist functional guild of ants was recorded.
Line 93: adding the region/country would strengthen this point also.
Line 99 to 100: Add "... community composition 'of ants'..." Is "identity of species present" not the same as species composition? If yes, I suggest removing the latter.
Can authors provide hypotheses that are supported by literature after the aim?

Methodology
Line 110: remove 'only', to read as "There are three resorts... "
Can you move sentences in Lines 119 to 128 to a sentence after Line 112 that talks about lack of permission to sample at Summer Bay resort? This way, you will be talking about your study areas before narrowing it to the site level.

I would remove the sentence in Line 112: "Due to the proximity of Sari Pacifica... "

Thanks to the authors for clarity on the sampling days. However, I suggest that the dates should match the number of days’ traps in the field, in that way exclude the day/s of traveling back to the mainland. For example, in Lines 133 to 136: ".... from 2nd to 7th of March.... from 26th September to 1st October..."

Line 138: "Pitfall traps were kept closed for 24 hours 'before opening' for sampling to reduce digging in effects.

Lines 138 to 139:" The traps were left open 'in the field' for five days.

Line 139: "On the 6th day of 'setting out traps', we dug up traps, packed it for transportation to the mainland."

Line 155: ".... sampling method was used" - I believe that authors used pitfall trapping only.

Line 162: How were the 20 traps arranged at each site?

Line 178: is Universiti not supposed to be University?

Line 210: In the past authors were italicizing citations but here italics are not used.

Line 217: "... of undisturbed sites 'yielded' the highest species

Line 218: "..... (Observed = 23 species, Estimated = 24-26 species), 'while' the pre-monsoon.... "

Line 231: "... 'the' Jaccard.... "

Line 241: Add Fig. 2 at the end of the sentence. "At disturbed sites, 17% of ant species were categorized as tramp/invasive species (Fig. 2). - I suggest removing the last part of the sentence.

Lines 247 to 249 seem to contradict Table 3. According to Table 3 GM had a single individual in undisturbed sites, yet this statement says there was dominance in undisturbed sites. Unless, authors meant disturbed sites. Also, I would say OPP were more in disturbed than undisturbed sites. SC is not found in disturbed sites but in undisturbed sites according to Table 3.

To make it less repetitive, I suggest that authors remove the first paragraph of the discussion (Lines 259 to 268). They can recap each results just before discussing it.

Line 269: What was the reasoning for high richness in undisturbed sites, it's probably common knowledge but can the authors spell it out.

Remove the sentence in Lines 272 to 273: Surprisingly, undisturbed habitats also harbored more tramp/invasive ant species (Fig. 2). These are results not discussion.

Line 274: Remove (Fig. 3)

Line 276: What is that dispersal range?

Lines 289, 299, and 304: Remove (Fig. 2) and (Table 2)

Line 322: indices not indexes.

Table 2: What does "others" refer to? Can this be defined in the caption.

---

## Round 0.3 · accepted · Accept

Dear Authors

Thank you for your submission of the revised version. The manuscript has been improved and found well for acceptance.